# Validation of the Arabic Maternal postpartum quality of life questionnaire among Lebanese women: A cohort study

**Mona Nabulsi**[1]*, **Hanan Smaili**[1], **Nour Abou Khalil**[2]

**1** Department of Pediatrics and Adolescent Medicine, American University of Beirut, Beirut, Lebanon,
**2** Faculty of Health Sciences, Department of Epidemiology and Population Health, American University of Beirut, Beirut, Lebanon

* mn04@aub.edu.lb

## Abstract

### Background

The postpartum period is an important phase in a woman's life. Yet, there is a paucity of validated instruments that assess maternal postpartum quality of life issues. The aim of this study is to describe the adaptation and validation of the Arabic version of the Maternal Postpartum Quality of Life (MAPP-QOL) questionnaire.

### Methods

This instrument validation cohort study tested an adapted Arabic version of the MAPP-QOL questionnaire on a convenience sample of 485 healthy Lebanese postpartum women. The MAPP-QOL reliability and validity were investigated by conducting Exploratory Factor Analysis using Principal Component Analysis, and by correlating the participants' MAPP-QOL scores with their scores on the Arabic Maternal Breastfeeding Evaluation Scale (MBFES-A), age, and education. Confirmatory Factor Analysis was conducted to examine how well the original factor structure of MAPP-QOL fits with our observed data using STATA 14. All other statistical analyses were done using SPSS version 23.

### Results

The Cronbach's alpha reliability coefficient of the Arabic MAPP-QOL was 0.90. Exploratory factor analysis revealed the following five components: *Functioning* (11 items, Cronbach's alpha of 0.82), *Socioeconomic* (9 items, Cronbach's alpha of 0.81), *Relational* (9 items, Cronbach's alpha of 0.75), *Psychological* (4 items, Cronbach's alpha of 0.74), and *Health* (6 items, Cronbach's alpha of 0.59). The overall Arabic MAPP-QOL score was positively but weakly correlated with the MBFES-A score ($r = 0.177$, $p < 0.001$), its Maternal Enjoyment/ Role Attainment subscale score ($r = 0.108$, $p = 0.023$), and Lifestyle/Body Image subscale score ($r = 0.286$, $p < 0.001$). There was no significant association between the type of infant feeding at one month and the Arabic MAPP-QOL score ($p = 0.932$). Similarly, the Arabic MAPP-QOL score was not correlated with the participant's age ($r = 0.043$, $p = 0.362$) or

**Data Availability Statement:** All relevant data are within the manuscript and its Supporting Information files.

**Funding:** This study was supported by a grant from the Medical Practice Plan, Faculty of Medicine, the American University of Beirut granted to MN. The funding body had no role in the design of the study, and collection, analysis, and interpretation of data, or in writing the manuscript.

**Competing interests:** The authors have declared that no competing interests exist.

**Abbreviations:** CFA, Confirmatory factor analysis; EFA, Exploratory factor analysis; IQR, Interquartile range; KMO, Kaiser-Meyer-Olkin; MBFES-A, Maternal Breastfeeding Evaluation Scale-Arabic; PCA, Principal component analysis; QLI, Quality of Life Index; QOL, Quality of life; MAPP-QOL, Maternal Postpartum Quality of Life Questionnaire; SD, Standard deviation.

education ($p = 0.451$). After modification of indices, Confirmatory Factor Analysis revealed that the goodness of fit indices corresponding to the 5-factor model in the original questionnaire indicate a reasonable fit with RMSEA = 0.052, CFI = 0.847 and SRMR = 0.062.

## Conclusions

The Arabic MAPP-QOL has good psychometric properties and may be a useful tool for clinicians and researchers interested in measuring maternal postpartum quality of life. Further replication of our findings in other Arab contexts is needed.

## Introduction

Ferrans and Powers defined quality of life (QOL) as "An individual's perceptions of well-being that stem from satisfaction or dissatisfaction with dimensions of life that are important to the individual" [1]. QOL is important to assess in all health conditions, especially in chronic pathologic conditions such as diabetes mellitus or cancer, or in physiologic conditions that normally persist for a relatively long time but can be demanding, such as the postpartum period in women. The postpartum period is challenging to most women as they must attend to several responsibilities simultaneously, like caring for the home, husband, children, and the newborn, as well as their careers, in case of maternal employment. This challenge may be augmented by physical stress post-delivery, such as pain from episiotomy or Cesarean surgical wounds, sleep deprivation, and breastfeeding. Moreover, some mothers may suffer from postpartum blues, or more serious problems such as postpartum depression, urinary or fecal incontinence, sexual problems, hemorrhoids, or breast problems [2–4].

Despite the importance of the postpartum period in a woman's life, and the different health problems that may arise during this period, there is a paucity of validated instruments to assess postpartum QOL. Hill, et al. were the first to develop and validate an instrument to specifically measure the QOL in postpartum women: the English Maternal Postpartum Quality of Life Questionnaire (MAPP-QOL) [5]. This instrument adopted the definition, domains, and conceptual model of the Quality of Life Index (QLI) of Ferrans and Powers [1]. The MAPP-QOL was initially tested on a cohort of American women and was proven to be a reliable and valid instrument with a Cronbach's alpha of 0.96. Subsequently, it was validated in Brazil and Iran, with both versions reporting similar Cronbach's alpha coefficients (Brazilian version: 0.88 [6]; Persian version: > 0.9) [7]). A different QOL instrument was recently developed by Zhou, et al. to specifically measure the QOL issues in Chinese settings but its reliability and validity awaits further testing [8].

There have been no studies from Lebanon that described the quality of life in postpartum women. Few studies from the Arab world assessed the QOL of postpartum Arab women and its association with different health-related issues or conditions [9–11]. A study from Saudi Arabia reported that 59.68% of postpartum women had probable postpartum depression, with an inverse correlation between postpartum depression and health-related quality of life scores measured by the Health-Related Quality of Life Scale [9]. In another cross-sectional study, Badr et al. recruited a convenience sample of Arabic-speaking postpartum women through Arab groups on Facebook and WhatsApp. They examined the associations between postpartum fatigue, depressive cognitions, life satisfaction, resourcefulness, and quality of life using the single-item global QOL question [10]. This study found significant correlations between postpartum fatigue and each of the three remaining variables. A third study from Egypt [11]

assessed the quality of life in postpartum women in relation to the various modes of delivery using the Chinese QOL by Zhou, et al., the reliability and validity of which were not tested [8]. There were no significant differences in the QOL aspects between mothers who delivered vaginally and those who delivered via Cesarean section. To note, none of the previous studies used the MAPP-QOL that is specific for the postpartum period. In view of the importance of the postpartum period in a woman's life, and its associated physical, emotional, and social changes, there is a need for a valid instrument that can reliably assess the postpartum QOL in Arab women. In this study, we explored whether an Arabic version of the MAPP-QOL is a valid and reliable instrument to measure the different aspects of postpartum QOL in a cohort of healthy Lebanese women.

## Materials and methods

### Design and setting

This study was approved by the Institutional Review Board of the American University of Beirut, reference number: PED.MN.16/SBS-2017-0450. Written informed consent was obtained from all participants.

Between April 2018 and February 2020, we recruited a total of 485 healthy Lebanese women. The inclusion criteria were: 1) delivery of a healthy term singleton newborn, 2) intention to breastfeed after delivery, 3) ability to read and write in Arabic. Exclusion criteria were: 1) women who had twins, 2) delivery before 37 weeks of gestation, 3) women diagnosed with a chronic disease, 4) not intending to breastfeed, 5) infant admission to the neonatal intensive care unit. Participants were consecutively recruited from the postpartum ward of a tertiary care academic center in the capital city Beirut. Most of the patients that seek health care at this center reside in Beirut and belong to the middle to high income strata of the population.

### Measurement

The Maternal Postpartum Quality of Life Questionnaire (MAPP-QOL) is a validated instrument that has two parts with identical items. The first part asks the participant about the extent of her *satisfaction* with a certain aspect of her life using a Likert-type scale that ranges from 1 'very dissatisfied' to 6 'very satisfied'. The second part asks about how *important* each item is to the participant using the same scale from 1 'very unimportant' to 6 'very important'. Scores are calculated by weighting each satisfaction response with its paired importance response, like the scoring of the QLI [1], with higher scores reflecting better quality of life. MAPP-QOL has 39 items divided over five domains: psychological/baby (8 items); socioeconomic (9 items); relational/spouse-partner (5 items); relational/family-friends (9 items); and health/functioning (8 items). The original MAPP-QOL has a Cronbach's alpha reliability coefficient of 0.96, with subscales' coefficients ranging between 0.82 and 0.96.

After obtaining the permission of the original author (P. D. Hill, written communication, November 16, 2012), the MAPP-QOL was translated to classical Arabic by one of the authors (MN), then back-translated to English by an independent translator who was blinded to the wording of the original English MAPP-QOL. We checked the back-translated English MAPP-QOL against the original English questionnaire and found them to be similar and accurate. The Arabic-translated MAPP-QOL was therefore piloted on 20 healthy mothers who delivered in the same center and presented for their one-month postpartum examination (n = 10), or the one-month well-child visit of their healthy newborns (n = 10) in the center's specialized clinics. All 20 participants found the questionnaire to be clear, easily understood, and culturally acceptable, but seven (35%) participants commented that it was long. The translated

Arabic MAPP-OQL was unchanged and administered in full to another cohort of participants for validation.

## Data collection

Trained research assistants approached eligible women in the privacy of their postpartum rooms to explain the study purpose and procedures and obtain their written informed consent. In addition to the Arabic-translated MAPP-QOL, information was collected on the following socio-demographics using a standardized questionnaire: the participant's age, household monthly income, highest attained education, parity, employment status, gestational length, number of breastfed children, longest duration of previous exclusive breastfeeding and any breastfeeding (for multiparous women), mode of delivery, newborn's gender, newborn's birth weight, and whether the participant had support at home. Information on the infants' nutrition was collected by telephone contact with the participants at two weeks, one month, and three months. Exclusive breastfeeding was defined as feeding the infant human milk only, with no other food or drink including water, but allowing oral rehydration solutions, vitamins, minerals, or other medicines when needed [12]. The participants were also administered the Arabic Maternal Breastfeeding Evaluation Scale (MBFES-A) at one month. This 26-item instrument has been shown to be a reliable and valid tool to measure maternal perceived overall quality of the breastfeeding experience in Lebanese women, with a Cronbach's alpha reliability coefficient of 0.87 [13]. The item scoring uses a 5-point Likert-type scale that ranges from 1 (*strong disagreement*) to 5 (*strong agreement*) resulting in a minimum score of 26 and a maximum of 130 points, with higher scores indicating more satisfaction with the breastfeeding experience.

The authors had access to information that would identify individual participants for the duration of the study. However, data were deidentified at study closure and after assuring that important data were not missing.

## Statistical analysis

The sample size calculation for this study was based on the recommended ten participants for every question in instrument validation studies [14,15]. Since the MAPP-QOL is composed of 39 items, we aimed to recruit at least 390 women. We inflated the sample size to 485 participants to account for a potential 25% loss to follow up. Data were summarized as means and standard deviations (*SD*) or medians and interquartile ranges (*IQR*) if continuous, and as frequencies and proportions if categorical. The Independent Student's *t* test or *ANOVA* were used to compare continuous variables, and the *Chi* Square test was used to compare categorical variables.

Participants' responses on the Arabic MAPP-QOL were scored in accordance with the scoring of the original English MAPP-QOL reported by Hill, et al. [5]. Briefly, scores were calculated by multiplying each satisfaction response by its paired importance response, centering the scale on 0 for satisfaction items. Then the resultant weighted items were summed, divided by the number of items answered, and negative values were eliminated by adding 15 to every score. Further details on the steps for calculating the scores are available on the QOL website by Ferrans [1].

Exploratory Factor Analysis (EFA) was conducted using Principal Component Analysis (PCA) with varimax rotation to assess its dimensionality and construct validity. The Kaiser-Meyer-Olkin (KMO) measure of sample adequacy and Bartlett's test of sphericity were done to test the suitability of the PCA method. The scree plot and the Eigenvalues were checked to decide on the number of factors that the items were loading on. The Arabic MAPP-QOL

internal consistency reliability was assessed using Cronbach's alpha coefficient and its item-total statistics such as item-total correlations and scale reliability coefficient if an item was deleted to decide which items to be retained.

Moreover, a confirmatory factor Analysis (CFA) was conducted using STATA 14 software to verify the factor structure for the MAPP-QOL hypothesized by Hill et al. [5]. The maximum-likelihood estimation method was employed, and a range of widely recognized goodness of fit indices were utilized to assess the adequacy of the model. The model fitness was assessed based on the Root Mean Square of Error of Approximation (RMSEA), Comparative Fit Index (CFI), Standardized Root Mean Square Residual (SRMR), and Chi-Square Statistic.

The predictive validity of MAPP-QOL was tested in bivariate analyses by correlating the participants' scores on MAPP-QOL with their MFBES-A scores, or with age using Pearson's correlation coefficient $r$, and with education using $t$ test. These three variables were chosen for this analysis because women who are more satisfied with their breastfeeding experience, are older in age, or are more educated may be more able to cope with the challenges of the postpartum period, and hence may have a better quality of life.

All analyses other than the CFA were done using SPSS version 23. Statistical significance was set at a $p$ value of $< 0.05$.

## Results

### Baseline characteristics

We recruited a total of 485 participants, of whom 24 withdrew by the second week, 11 withdrew between two weeks and one month, and 2 withdrew between one and three months, with an overall loss to follow up of 7.6% (Fig 1).

The participants had a mean (*SD*) age of 31.1 (4.9) years and had completed a mean (*SD*) of 38.4 (1.3) weeks of pregnancy. About half of them were primiparous ($n = 218$; 44.9%), with 201 women (41.4%) delivering by Cesarean delivery. The majority had university education ($n = 421$; 86.8%), 421 (86.8%) mothers were employed with 84.0% of them having a full-time job. Table 1 summarizes the participants' remaining baseline characteristics.

### Reliability

The Cronbach's alpha internal consistency reliability of the Arabic MAPP-QOL was 0.90. The scree plot that was generated from EFA (Fig 2) suggested that the scale had five component loadings with a total variance of 42.6%. Table 2 reveals the Eigen values and the total variance for the five components.

The KMO measure of sample adequacy was 0.864 ($p < 0.001$) implying adequacy of the PCA [16]. The items included in each component are shown in Table 3.

The labeling of the components/subscales was modified from the original English MAPP-QOL because several items relocated to subscales different from those reported in the original English questionnaire as shown in Table 3. The components/subscales were named as follows: *Functioning* (11 items, Cronbach's alpha of 0.82), *Socioeconomic* (9 items, Cronbach's alpha of 0.81), *Relational* (9 items, Cronbach's alpha of 0.75), *Psychological* (4 items, Cronbach's alpha of 0.74), and *Health* (6 items, Cronbach's alpha of 0.59). Item 17 (*Your baby's health*) loaded equally on *Relational* and on *Health* subscales. We decided therefore to keep it in the *Health* subscale because it is asking about the baby's health. There were three items with coefficients below 0.3. These were item 7 (*Your breasts*), item 19 (*Time for children*), and item 34b (*Your own employment*). We decided to keep these three items because of their importance to maternal quality of life. The participants' overall MAPP-QOL score ranged between 10.0 and 30.0, with a mean (SD) of 23.3 (3.1).

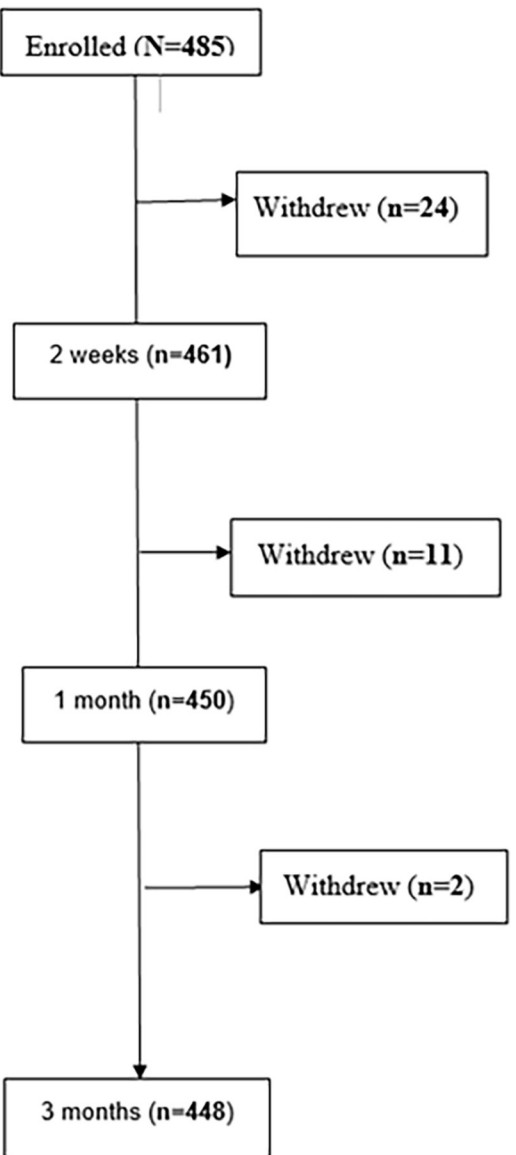

**Fig 1. Flow diagram.** Participants' flow through the study. The plot shows the eigenvalues, usually displaying a curve with an "elbow" shape, with the number of points above the "elbow" considered as an estimate of the number of factors to retain.The plot shows the eigenvalues, usually displaying a curve with an "elbow" shape, with the number of points above the "elbow" considered as an estimate of the number of factors to retain.The plot shows the eigenvalues, usually displaying a curve with an "elbow" shape, with the number of points above the "elbow" considered as an estimate of the number of factors to retain.

In the CFA, after modification of indices, the goodness of fit indices corresponding to the 5-factor model suggested in the original questionnaire, indicated a reasonable fit with RMSEA = 0.052, CFI = 0.847 and SRMR = 0.062. We failed to reproduce the covariance matrix with a *p*-value < 0.0001. Additionally, all items showed significant factor loadings on their respective factors. The lowest factor loading was observed for item 8 "*Your surgical incision or episiotomy*", whereas the highest was observed for item 30 "*Your ability to meet financial obligations*".

**Table 1. Participants' baseline characteristics (N = 485).**

| Continuous Variables (normal distribution) | Mean (*SD*) |
|---|---|
| Infant's birth weight (Grams) | 3265.6 (405.1) |
| MBFES-A overall score‡ | 104.5 (12.9) |
| Subscale Infant Satisfaction/Growth score‡ | 38.0 (5.7) |
| Subscale Maternal Enjoyment/Role Attainment score‡ | 47.6 (6.0) |
| Subscale Lifestyle/Body Image score‡ | 18.9 (4.7) |
| **Continuous Variables (skewed distribution)** | **Median (*IQR*)** |
| Number of children† <br> Range | 1 (1, 2) <br> 1 to 5 |
| Number of breastfed children† <br> Range | 1 (1, 2) <br> 0 to 5 |
| Longest duration of previous EBF†‡ (months) <br> Range | 3 (0.0, 11.8) <br> 0.0 to 29.0 |
| Longest duration of previous mixed feeding†‡ (months) <br> Range | (0.0–5.0) <br> 0.0 to 30.0 |
| **Categorical Variables** | ***n* (%)** |
| Male infant | 231 (47.6) |
| Employed, *n* (%) <br> Can leave work to BF <br> Can pump at work | 293 (60.4) <br> 99 (33.8) <br> 243 (82.9) |
| Monthly income ($) <br> ≤ 1000 <br> 1001 - <5000 <br> ≥ 5000 <br> Missing | <br> 63 (13.0) <br> 304 (62.7) <br> 110 (22.7) <br> 8 (1.6) |
| Has support at home | 470 (96.9%) |
| Infant nutrition at 2 weeks <br> EBF <br> Mixed feeding <br> Artificial milk <br> Missing | <br> 245 (50.5) <br> 186 (38.4) <br> 30 (6.2) <br> 24 (4.9) |
| Infant nutrition at 1 month <br> EBF <br> Mixed feeding <br> Artificial milk <br> Missing | <br> 209 (43.1) <br> 190 (39.2) <br> 51 (10.5) <br> 35 (7.2) |
| Infant nutrition at 3 months <br> EBF <br> Mixed feeding <br> Artificial milk <br> Missing | <br> 168 (34.6) <br> 150 (30.9) <br> 130 (26.8) <br> 37 (7.7) |

<u>Note:</u> EBF = exclusive breastfeeding, †For multiparous participants, ‡Missing values: MBFES-A scale and subscale scores = 35, longest duration of EBF = 23, longest duration of mixed breastfeeding = 23.

## External validity

The bivariate analysis revealed no significant differences in the mean (SD) overall Arabic MAPP-QOL score and the type of infant feeding. The score was 23.2 (3.1) for participants who were exclusively breastfeeding at one month, 23.4 (3.2) for mothers whose infants were on mixed feeding, and 23.3 (2.9) for those whose infants were exclusively on artificial milk.

The overall Arabic MAPP-QOL score was positively but weakly correlated with the overall MBFES-A score ($r = 0.177$, $p < 0.001$), the Maternal Enjoyment/Role Attainment MBFES-A subscale score ($r = 0.108$, $p = 0.023$), and lifestyle/Body Image MBFES-A subscale score

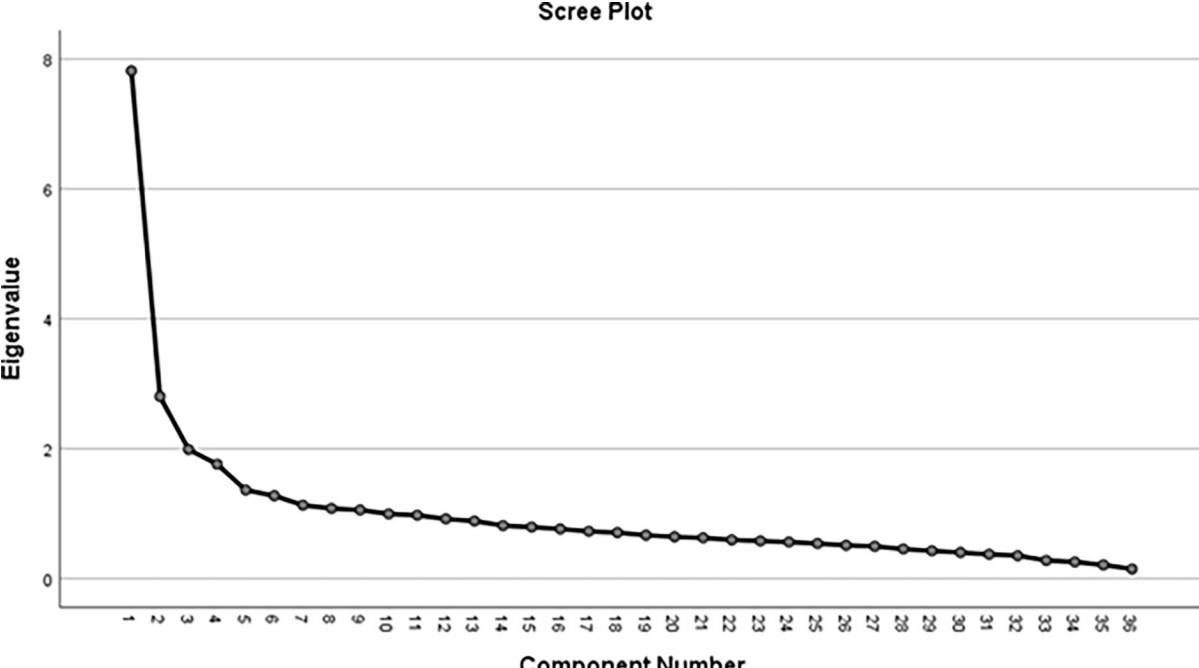

**Fig 2. Scree plot of the Arabic MAPP-QOL.** The first five points above the curve's "elbow" have Eigen values above 1.The plot shows the eigenvalues, usually displaying a curve with an "elbow" shape, with the number of points above the "elbow" considered as an estimate of the number of factors to retain.The plot shows the eigenvalues, usually displaying a curve with an "elbow" shape, with the number of points above the "elbow" considered as an estimate of the number of factors to retain.The plot shows the eigenvalues, usually displaying a curve with an "elbow" shape, with the number of points above the "elbow" considered as an estimate of the number of factors to retain.

($r = 0.286$, $p < 0.001$). However, there were no significant correlations with the Infant Satisfaction/Growth MBFES-A subscale score ($r = 0.051$, $p = 0.286$), participant's age ($r = 0.043$, $p = 0.362$), or education ($p = 0.451$).

## Discussion

Despite of its importance, the assessment of maternal quality of life in the early postpartum period is currently not a routine obstetric practice. Our literature review revealed that there is only one QOL instrument, other than the MAPP-QOL, that was specifically developed to measure postpartum QOL [8]. This Chinese instrument however is limited by the fact that the authors did not report its reliability and validity estimates. Moreover, few studies from the Arab world investigated the QOL of postpartum women and its association with different health conditions, however the instruments used were not specific for the postpartum period

**Table 2. Eigen values for the 5 components of the scale and the total variance.**

| Component | Initial Eigen Values | | |
|---|---|---|---|
| | Total | % of Variance | Cumulative % |
| 1 | 8.570 | 21.975 | 21.975 |
| 2 | 2.878 | 7.380 | 29.355 |
| 3 | 1.998 | 5.123 | 34.477 |
| 4 | 1.807 | 4.633 | 39.111 |
| 5 | 1.374 | 3.523 | 42.633 |

**Table 3. Factor loadings for the 5-components and communalities of the Arabic MAPP-QOL with all 39 items.**

| Scale and items | Factors | | | | | Communalities |
|---|---|---|---|---|---|---|
| | **1** | **2** | **3** | **4** | **5** | |
| *1. Functioning* | | | | | | |
| **Q23.Time for yourself** | **0.715** | | | | | 0.542 |
| **Q4. Amount of control you have over your life** | **0.650** | | | | | 0.472 |
| **Q16. Your ability to meet family responsibilities** | **0.642** | | | | | 0.482 |
| **Q21. Time for friends & relatives** | **0.600** | | | | | 0.396 |
| **Q20. Time for maintaining the household** | **0.582** | | | | | 0.374 |
| Q3. Amount of energy for every day activities | **0.571** | | | | | 0.472 |
| **Q22. Time for husband** | **0.549** | | 0.428 | | | 0.513 |
| Q6. Your physical appearance | **0.536** | | | | | 0.383 |
| **Q26. Your day-to-day life routine** | **0.517** | | 0.308 | | | 0.399 |
| Q5. Your ability to take care of yourself without help | **0.463** | | | | | 0.250 |
| Q9. Your sex life | **0.355** | | | | | 0.235 |
| *2. Socioeconomic* | | | | | | |
| Q33b. Your economic or financial capacity | | **0.826** | | | | 0.733 |
| Q30. Your ability to meet financial obligations | | **0.755** | | | | 0.625 |
| Q33a. Your materialistic possessions | | **0.744** | | | | 0.610 |
| Q29. Your financial independence | | **0.722** | | | | 0.583 |
| Q32. Your access to transportation | | **0.501** | | | | 0.290 |
| Q28. Your neighborhood | | **0.469** | | | | 0.338 |
| Q31. Your access to medical care | | **0.466** | 0.353 | | | 0.433 |
| Q27. Your home/appartment/place where you live | | **0.354** | 0.317 | | | 0.295 |
| Q34b. Your own employment | | **0.315** | | | | 0.198 |
| *3. Relational* | | | | | | |
| Q15. Your relationship with your husband | | | **0.782** | | | 0.653 |
| Q14a. Emotional support you get from your husband | | | **0.676** | | | 0.499 |
| Q14b. Emotional support you get from your extended family | | | **0.558** | | | 0.374 |
| Q25. Your husband's health | | | **0.500** | | | 0.343 |
| Q34a. Your husband's employment | | 0.373 | **0.499** | | | 0.411 |
| Q33c. Your overall environment/surroundings | | 0.316 | **0.452** | | | 0.312 |
| Q14c. Emotional support you get from your friends or other people | | | **0.389** | | | 0.312 |
| Q18. Assistance with baby care & other children | | | **0.378** | | | 0.262 |
| Q19. Time for children | | | **0.316** | | | 0.205 |
| *4. Psychological* | | | | | | |
| Q11. Your happiness in general | | | 0.328 | **0.792** | | 0.772 |
| Q12. Your life in general | | | | **0.775** | | 0.719 |
| Q10. Your peace of mind | 0.332 | | | **0.515** | | 0.479 |
| Q13. Your amount of worries in life | | | | **0.310** | | 0.301 |
| *5. Health* | | | | | | |
| Q8. Your surgical incision or episiotomy | | | | | **0.588** | **0.380** |
| Q2. Amount of pain that you have | | | | 0.404 | **0.576** | **0.508** |
| Q7. Your breasts | | | | | **0.501** | **0.338** |
| Q1. Your health | | | | | **0.471** | **0.368** |

*(Continued)*

**Table 3.** (Continued)

| Scale and items | Factors | | | | | Communalities |
|---|---|---|---|---|---|---|
| | 1 | 2 | 3 | 4 | 5 | |
| Q24. **Your ability to feed your new baby** | | | 0.301 | | **0.455** | **0.383** |
| Q17. **Your baby's health** | | | 0.404 | | **0.404** | **0.386** |

<u>Note</u>: Bold items are items that relocated from other subscales of the English MAPP-QOL.

[9,10]. In this study, we validated a much-needed Arabic version of the MAPP-QOL questionnaire that can be used in maternal health research, as well as in clinical practice.

The Arabic MAPP-QOL and its subscales have comparable reliability coefficients to the original English MAPP-QOL questionnaire [5]. The main differences between the English and the Arabic MAPP-QOL versions were the relocations of some items in the English questionnaire from one subscale to a different one in the Arabic version (Table 3). For example, the eight items in the English *Health and Functioning* subscale were subdivided in the Arabic version between the *Health* subscale (4 items: Your surgical incision or episiotomy; Amount of pain that you have; Your breasts; Your health), and the *Functioning* subscale (4 items: Amount of energy for every day activities; Your physical appearance; Your ability to take care of yourself without help; Your sex life). This division into separate subscales in the Arabic version is more appropriate for our setting since the *Health* items are quite different from the *Functioning* items and fit more with the names of these subscales. Two other items (Your baby's health; Your ability to feed your new baby) relocated from the English *Psychological/Baby* subscale to the Arabic *Health* subscale. These two items are also assessing infant health aspects that fit better with the *Health* subscale. Similarly, seven items relocated to the Arabic *Functioning* subscale: five from the English *Relational/family-friends* subscale (Time for yourself; Your ability to meet family responsibilities; Time for friends and family; Time for maintaining the household; Time for husband), and two from the English *Psychological/Baby* subscale (Amount of control you have over your life; Your day-to-day life routine). All these relocated items are about life aspects that in our context are related to an individual's daily function. The Arabic *Socioeconomic* subscale retained the same items of the English version, whereas the *Relational* subscale retained five items from the English *Relational/husband-partner* and four items from the *Relational/family-friends* subscales (Table 3).

In our cohort, mothers who had higher scores on the Arabic-MAPP-QOL suggesting better quality of life were also more satisfied with their overall breastfeeding experience, maternal enjoyment and role attainment, and lifestyle and body image. These findings are in line with previous literature reported by other investigators. In a study from Spain, Triviño-Juárez et al. reported that breastfeeding was associated with greater maternal quality of life as measured by the Spanish SF-36 questionnaire [17]. This finding was attributed to having a higher proportion of breastfed infants who ate and slept well and had less emergency room illness visits. Moreover, there is evidence of an association between breastfeeding and body image. A systematic review that examined the association between body image and breastfeeding initiation and duration reported that breastfeeding was less likely initiated and was shorter in duration in mothers with body concerns as compared to mothers with a higher body image [18].

The modest correlation between maternal quality of life and satisfaction with certain aspects of breastfeeding domains supports the external validity of the Arabic MAPP-QOL. However, satisfaction with breastfeeding did not translate into higher rates of exclusive breastfeeding in this cohort, a finding that is consistent with our previous report on the same cohort [13]. It is well-established that a mother's decision to breastfeed her infant and the duration of

breastfeeding may be affected by several factors, and not only her satisfaction with the breastfeeding experience. In a previous study from Lebanon, we identified several barriers and challenges as reasons for early breastfeeding discontinuation. These included negative body image (breast sagging or weight gain), pain, sleep deprivation, exhaustion, lack of support, and maternal employment [19]. To note, in the English MAPP-QOL validation study, only 78 of the 184 participants (42.4%) chose to provide mother's milk, 43 (23.4%) chose formula, and 63 (34.2%) elected to provide both mother's milk and formula. The authors did not investigate whether an association existed between the type of infant nutrition and the participants' MAPP-QOL scores. Similarly, both the Brazilian and Persian maternal postpartum quality of life studies did not report on any association between breastfeeding and maternal quality of life [5–7]. Hence, our study is the first one to explore such an association in a large sample of postpartum women. Moreover, our study has the largest sample size when compared to the English, Brazilian, or Persian MAPP-QOL validation studies [5–7]. In our attempt to examine the fit of the original factor structure suggested by Hill et al [5] using CFA, the model seemed to approximate the data reasonably, yet it did not demonstrate perfect alignment with our observed data given the unique context of our study.

Our study's main limitation is that most of the participants were recruited from a major tertiary care center which serves highly educated, middle- or high-income women from the capital city. Therefore, the findings of this study may not be generalizable to other Lebanese women with different socioeconomic or education status, or to other Arab women.

## Conclusions

The Arabic MAPP-QOL has good psychometric properties. This study provides the first Arabic instrument that can be used by clinicians to measure the QOL of Arab postpartum women, a period with relatively increased prevalence of significant social stressors and physical ailments that can negatively affect their QOL. Moreover, this valid instrument may bolster research efforts focusing on QOL issues in the Arab world. Future research is needed to replicate our findings in other Arab contexts, as well as to further refine the instrument for optimal fit with our cultural and contextual variations.

## Supporting information

**S1 File. The Arabic MAPP-QOL.**
(PDF)

**S2 File. Minimal dataset.**
(XLSX)

## Acknowledgments

We are thankful to all our participants for their valuable contribution to our study.

## Author Contributions

**Conceptualization:** Mona Nabulsi.

**Data curation:** Mona Nabulsi, Hanan Smaili.

**Formal analysis:** Mona Nabulsi, Nour Abou Khalil.

**Funding acquisition:** Mona Nabulsi.

**Investigation:** Mona Nabulsi.

**Methodology:** Mona Nabulsi.

**Project administration:** Mona Nabulsi, Hanan Smaili.

**Resources:** Mona Nabulsi.

**Software:** Mona Nabulsi.

**Supervision:** Mona Nabulsi.

**Validation:** Mona Nabulsi, Nour Abou Khalil.

**Visualization:** Mona Nabulsi.

**Writing – original draft:** Mona Nabulsi.

**Writing – review & editing:** Mona Nabulsi, Hanan Smaili, Nour Abou Khalil.

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
