## [Decision Letter · Decision Letter 0]

30 May 2023

PONE-D-23-06864Validation of the Arabic Maternal Postpartum Quality of Life Questionnaire (MAPP-QOL) Among Lebanese Women: A cohort studyPLOS ONE

Dear Dr. Nabulsi, Thank you for submitting your manuscript to PLOS ONE. After careful consideration, we feel that it has merit but does not fully meet PLOS ONE’s publication criteria as it currently stands. Therefore, we invite you to submit a revised version of the manuscript that addresses the points raised during the review process.

We look forward to receiving your revised manuscript.

Kind regards,

Othman A. Alfuqaha, Ph.D.

Academic Editor

PLOS ONE

Journal Requirements:

Additional Editor Comments:

Dear authors, regarding your paper of validation of the Arabic MAPP-QOL Among Lebanese Women. Please respond to all reviewers' comments and remove the abbreviation from the title. We hope that the authors promptly address all comments and submit their revised paper, allowing me to continue the review process. If you require any further assistance or have any specific concerns regarding the paper, please do not hesitate to let us know.

Cordially

Reviewers' comments:

Reviewer's Responses to Questions

**Comments to the Author**

1. Is the manuscript technically sound, and do the data support the conclusions?

Reviewer #1: Partly

Reviewer #2: Partly

Reviewer #3: Yes

2. Has the statistical analysis been performed appropriately and rigorously? 

Reviewer #1: Yes

Reviewer #2: Yes

Reviewer #3: Yes

3. Have the authors made all data underlying the findings in their manuscript fully available?

Reviewer #1: Yes

Reviewer #2: Yes

Reviewer #3: Yes

4. Is the manuscript presented in an intelligible fashion and written in standard English?

Reviewer #1: Yes

Reviewer #2: Yes

Reviewer #3: Yes

5. Review Comments to the Author

Reviewer #1: 1. The sample size calculation needs to be specified.

2. The statistical analysis section is limited to exploratory factor analysis only. The domains need to be rechecked using confirmatory factor analysis. Please add.

3. A study flow needs to be added.

Reviewer #2: ID: PONE-D-23-06864

Title: Validation of the Arabic Maternal Postpartum Quality of Life Questionnaire (MAPP-QOL) Among Lebanese Women

Thank you for providing a chance to review this manuscript.

Comment: Major Revision.

Detailed information:

ABSTRACT

Results: Where are the validity test results reflected? Please add.

Methods: What method was used to select the sample? What methods and software are used for statistical analysis?

BACKGROUND

Line 88-90, page 5: Just because Lebanon does not have an Arab version, so do it? Is this the so-called research meaning? Please add clarification.

Overall: 1) What was the basis for your topic? Could some research background be added on the quality of life of postpartum women in Lebanon or Arabia? 2) What is your research hypothesis？

METHODS

Design and setting:

Line 97-100, page 5: The description of the inclusion criteria should be more complete and clarifications such as "firstly" could be considered. Similarly, the exclusion criteria should be clearly written.

Line 101-102, page 5: 1) Is it appropriate to select the sample only from a tertiary nursing academic center in the capital city Beirut? Can the results of the study be extrapolated? I think it is under-represented. 2) Please add the relevant content of the ethical review.

Measurement

Line 101-111, page 5-6: Most of this is duplicated with the background part, please modify it.

Data collection

Line 145, page 7: “Arabic Maternal Breastfeeding Evaluation Scale (MBFES-A)” how reliable is the scale valid? Please provide a brief description of the entry and score for the Maternal Breastfeeding Evaluation Scale (MBFES-A).

Statistical Analysis

Line 160, page 8: You only used exploratory factor analysis, what about confirmatory factor analysis?

Line 168, page 8: The research hypothesis should be placed in the last paragraph of the introduction. The Statistical Analysis section states only the statistical analysis method.

RESULTS

Line 117, page 8: Since 39 of the 485 people recruited dropped out, shouldn't your final sample size be 446? What exactly is the value of "N" in the header of the table? Please confirm.

Table 1, page 9: Please make sure that the sum of the components in the table is 100%.

Line 203, page 12: The Cronbach coefficient for the health dimension is less than 0.7, how do you solve it?

Overall: 1) If there are no special requirements, it is recommended to use a tri-line table for all forms to make it more aesthetically pleasing. 2) The results section can list sub-headings for each section separately to make the logic clearer.

DISCUSSION

Overall: 1) The discussion section has too few references, please add relevant arguments. 2) The discussion section did not explain the interpretation of the results and subsequent processing methods. Please fully explain and compare the results section.

CONCLUSIONS

Line 265-266, page 14: So what is the relationship between the quality of life of women in the early postpartum period?

In my opinion, the starting point and research theme of this article are innovative. However, the logic and English formulation in the article need to be strengthened. A major question is, why is confirmatory factor analysis not directly used for QOLs with relatively stable structures? In addition, the discussion section did not explain the results reasonably and was confusing. It is recommended to read more references to provide theoretical support. Lastly, the typography of the text of the whole article is particularly messy, please pay attention to the alignment.

Thank you and my best,

Your reviewer

Reviewer #3: Dear Authors,

This is a very well written and valuable paper. I have some minor suggestions.

- The method of sampling should be added in method portion of manuscript.

- The communalities of all items need to be provided alongside the factor loads.

- Please include Arabic version of the MAPP-QOL score as a supplementary material for the accessible use of other researchers and clinicians.

- Why confirmatory factor analysis (CFA) has not been used?

- have the face validity and content validity been performed after the forward-backward translation?

- Although the slope on the scree plot was leveling after fourth factor, it seems that the decision to retain five factors was to some extent arbitrary. Additional arguments for such a decision could be obtained by employing some further procedures aimed at determining the optimal number of factors, such as Horn's parallel analysis or Velicer's minimum average partial.

Best Regards,

6. PLOS authors have the option to publish the peer review history of their article (what does this mean?). If published, this will include your full peer review and any attached files.

Reviewer #1: **Yes: **Subhranil Saha

Reviewer #2: No

Reviewer #3: No

---

## [Author Response · Author response to Decision Letter 0]

3 Jul 2023

Dear Editor,

We would like to submit our revised manuscript [PONE-D-23-06864] entitled “Validation of the Arabic Maternal Postpartum Quality of Life Questionnaire Among Lebanese women: A Cohort Study” for publication in PLOS ONE. All changes have been highlighted in yellow in the revised tracked copy. We are very thankful for the constructive comments of the reviewers which we hope will improve our manuscript. 

Below, please find our response to each comment.

Journal Requirements:

1. Please ensure that your manuscript meets PLOS ONE’s style requirements, including those for file naming.

Answer: Done. The manuscript was revised in accordance with PLOS ONE’s style requirements, including file naming. 

2. Minimal data set: Please upload your study’s minimal data set as either Supporting Information files or to a stable public repository.

Answer: We uploaded our minimal deidentified data set as Supporting Information file (S2 File). 

3. Data Availability: Please describe the changes you make to Data Availability Statement in your cover letter.

Answer: We submitted our anonymized dataset to PLOS ONE with the revised manuscript as Supporting Information File (S2 File). Kindly update our Data Availability statement on our behalf to reflect this information.

4. Your Ethics statement should only appear in the Methods section of your manuscript. Please ensure that your ethics statement is included in your manuscript. 

Answer: We included the following ethics statement in the Methods section: 

“This study was approved by the Institutional Review Board of the American University of Beirut, reference number: PED.MN.16/SBS-2017-0450. Written informed consent was obtained from all participants.”

Editor’s comments:

- Please remove the abbreviation from the title.

Answer: We removed the abbreviation from the title.

Response to Reviewer 1: 

Thank you for your constructive comments and the valuable review that helped us improve our manuscript. Please find below our reply to each comment.

1. The sample size calculation needs to be specified. 

Answer: We added the following statement and references to the Methods/Statistical analysis section: 

“The sample size calculation for this study was based on the recommended ten participants for every question in instrument validation studies [14, 15]. Since the MAPP-QOL is composed of 39 items, we aimed to recruit at least 390 women. We inflated the sample size to 485 participants to account for a potential 25% loss to follow up.” 

2. The statistical analysis section is limited to exploratory analysis only. The domains need to be rechecked using confirmatory factor analysis. Please add. 

Answer: Thank you for your comment. The reason we conducted Exploratory Factor Analysis is because the original English MAPP-QOL was validated in only 2 contexts (Brazilian and Persian) that are distinctly different from the Lebanese context. Hence, we wanted to explore how it would perform in our culture. Confirmatory Factor Analysis would be more suitable when the instrument has been well established to have good reliability and validity in most settings (e.g. The Beck Depression Inventory used to screen for depression and its severity in people over 13 years of age). 

3. A study flow needs to be added.

Answer: We added a flow diagram (Figure 1). 

4. ABSTRACT/Results: Where are the validity test results reflected? Please add. 

Answer: Kindly refer to the below paragraph in the revised Abstract/Results: 

“The overall Arabic MAPP-QOL score was positively but weakly correlated with the Arabic Maternal Breastfeeding Evaluation Scale score (r = 0.177, p < 0.001), its Maternal Enjoyment/Role Attainment subscale score (r = 0.108, p = 0.023), and Lifestyle/Body Image subscale score (r = 0.286, p < 0.001). There was no significant association between the type of infant feeding at one month and the Arabic MAPP-QOL score (p = 0.932). Similarly, the Arabic MAPP-QOL score was not correlated with the participant’s age (r = 0.043, p = 0.362) or education (p = 0.451)”

5. ABSTRACT/Methods: What method was used to select the sample? What methods and software were used for statistical analysis? 

Answer: We revised the Abstract/Methods to describe how the sample was selected, and summarized the methods used in statistical analysis and the software as follows:

“This instrument validation cohort study tested an adapted Arabic version of the MAPP-QOL questionnaire on a convenience sample of 485 healthy Lebanese postpartum women. The MAPP-QOL reliability and validity were investigated by conducting Exploratory Factor Analysis using Principal Component Analysis, and by correlating the participants’ MAPP-QOL scores with their scores on the Arabic Maternal Breastfeeding Evaluation Scale (MBFES-A), age, and education. All statistical analyses were done using SPSS version 23” 

6. BACKGROUND: Line 88-90, page 5: Just because Lebanon does not have an Arab version, so do it? Is this the so-called research meaning? Please add clarification. 

Answer: Thank you for your comment. We added the following paragraph to the Background of the revised manuscript clarifying the existing knowledge gap and the need for an Arabic version of the MAPP-QOL:

“There are few studies from the Arab world that assessed the QOL of Arab women during the postpartum period and its association with different health-related issues or conditions. None of these used the MAPP-QOL that is specific for this period [9-10]. The QOL instruments used in these studies included the Health-Related Quality of Life Scale [9], the single-item global QOL question [10], and the Chinese QOL instrument by Zhou, et al. the reliability and validity of which were not tested [8, 11]. Hence, there is a need for a valid instrument that can reliably assess the postpartum QOL in Arab women. In this study, we explored whether an Arabic version of the MAPP-QOL is a valid and reliable instrument to measure the different aspects of postpartum QOL in a cohort of healthy Lebanese women.” 

7. BACKGROUND/Overall: 1) What was the basis of your topic? Could some research background be added on the quality of life of postpartum women in Lebanon or Arabia? 2) What is your research hypothesis? 

Answer: We addressed your comment about the basis of our topic in the following paragraph (similar to comment 6 above):

“There are few studies from the Arab world that assessed the QOL of Arab women during the postpartum period and its association with different health-related issues or conditions. None of these used the MAPP-QOL that is specific for this period [9-10]. The QOL instruments used in these studies included the Health-Related Quality of Life Scale [9], the single-item global QOL question [10], and the Chinese QOL instrument by by Zhou, et al. the reliability and validity of which were not tested [8, 11]. Hence, there is a need for a valid instrument that can reliably assess the postpartum QOL in Arab women.” 

Our research hypothesis is summarized in the below statement:

In this study, we explored whether an Arabic version of the MAPP-QOL is a valid and reliable instrument to measure the different aspects of postpartum QOL in a cohort of healthy Lebanese women.” 

8. METHODS/Design and setting: Line 97-100, page 5: The description of the inclusion criteria should be more complete and clarifications such as “firstly” could be considered. Similarly, the exclusion criteria should be clearly written. 

Answer: We revised the statement on inclusion and exclusion criteria as suggested:

“The inclusion criteria were: 1) delivery of a healthy term singleton newborn, 2) intention to breastfeed after delivery, 3) ability to read and write in Arabic. Exclusion criteria were: 1) women who had twins, 2) delivery before 37 weeks of gestation, 3) women diagnosed with a chronic disease, 4) not intending to breastfeed, 5) infant admission to the neonatal intensive care unit” 

9. METHODS/Design and setting: Line 101-102, page 5: 1) Is it appropriate to select the sample only from a tertiary academic center in the capital city Beirut? Can the results of the study be extrapolated? I think it is under-represented. 2) Please add the relevant content of the ethical review.

Answer: We agree with the kind reviewer that our sample is not representative of all Lebanese women. However, with the resources available to us, we could only recruit from our center which is the largest referrral academic center in Lebanon. We have addressed this limitation in the Discussion section as follows: 

“Our study’s main limitation is that most of the participants were recruited from a major tertiary care center which serves highly educated, middle- or high-income women from the capital city. Therefore, the findings of this study may not be generalizable to other Lebanese women with different socioeconomic or education status, or to other Arab women” 

We also added the following statement on the ethical review:

“This study was approved by the Institutional Review Board of the American University of Beirut, reference number: PED.MN.16/SBS-2017-0450. Written informed consent was obtained from all participants.”

10. METHODS/Measurement: Line 101-111, page 5-6: Most of this is duplicated with the background part. Please modify it.

Answer: Thank you. We removed the redundant statements from the Measurement section of the revised manuscript as suggested.

11. METHODS/Data collection: Line 145, page 7: “Arabic Maternal Breastfeeding Scale (MBFES-A)” How reliable or valid is the scale? Please provide a brief description of the entry and score for the MBFES-A.

Answer: We added the following statements that describe the reliability and validity of the MFFES-A, and its scoring:

“The participants were also administered the Arabic Maternal Breastfeeding Evaluation Scale (MBFES-A) at one month. This 26-item instrument has been shown to be a reliable and valid tool to measure maternal perceived overall quality of the breastfeeding experience in Lebanese women, with a Cronbach’s alpha reliability coefficient of 0.87 [13]. The item scoring uses a 5-point Likert-type scale that ranges from 1 (strong disagreement) to 5 (strong agreement) resulting in a minimum score of 26 and a maximum of 130 points, with higher scores indicating more satisfaction with the breastfeeding experience.”

12. METHODS/Statistical analysis: Line 160, page 8: You only used exploratory factor analysis, what about confirmatory factor analysis?

Answer: We addressed this comment in our reply to comment 2 above. 

13. METHODS/Statistical analysis: Line 168, page 8: The research hypothesis should be placed in the last paragraph of the introduction. The statistical analysis section states only the statistical analysis method. 

Answer: We agree with the kind reviewer about this point. Kindly note that the aim of this study is to explore whether the MAPP-QOL is reliable and valid in our context. This is now clearly stated in the last paragraph of the Background section. 

“In this study, we explored whether an Arabic version of the MAPP-QOL is a valid and reliable instrument to measure the different aspects of postpartum QOL in a cohort of healthy Lebanese women.” 

We revised the Statistical section so the reader is not confused about the research hypothesis regarding the use of the variables MBFES-A scores, participant age and education as follows: 

“The predictive validity of MAPP-QOL was tested in bivariate analyses by correlating the participants’ scores on MAPP-QOL with their MFBES-A scores, or with age using Pearson’s correlation coefficient r, and with education using t test. These three variables were chosen for this analysis because women who are more satisfied with their breastfeeding experience, are older in age, or are more educated may be more able to cope with the challenges of the postpartum period, and hence may have a better quality of life” 

14. RESULTS: Line 117, page 8, Since 39 of the 485 people recruited dropped out, shouldn’t your final sample size be 446? What exactly is the value of “N” in the header of the Table? Please confirm.

Answer: I believe the kind reviewer is referring to Table 1 which summarizes the baseline characteristics of the 485 recruited participants. In fact, the withdrawals were 37 and not 39 women. We corrected this error in the revised manuscript. 

The reason we wrote N=485 is because we have data on their baseline characteristics at enrolment, and the 37 withdrawals happened later during the study. It is usually expected of authors to report all available data at each time point of the study.

“N” refers to the total sample size, whereas ‘n’ refers to a subgroup of the total sample.

15. RESULTS: Table 1, page 9: Please make sure that the sum of the components in the table is 100%.

Answer: Thank you for alerting us to this point. We checked all values and added the missing values so that the sum of the components is 100%. 

16. RESULTS: Line 203, page 12: The Cronbach coefficient for the health dimension is less than 0.7, how do you solve it?

Answer: Despite the low Cronbach coefficient of the Health dimension, we decided to keep this dimension for several reasons: 1) The health dimension and the items included in it are all health aspects that are important for postpartum women (e.g. wound, episiotomy, pain, etc..); 2) All items in this dimension have communalities ranging between 0.338 and 0.580; 3) When we tried different scenarios in which items were deleted and the Factor loading was repeated, we got lower Cronbach alpha values. We even tried the scenario of 4 components instead of 5 and also got lower reliability. Hence, we decided on keeping this dimension despite its low Cronbach coefficient because of the importance of its items to QOL in the postpartum period. 

17. RESULTS/Overall: 1) If there are no special requirements, it is recommended to use a tri-line table for all forms to make it more aesthetically pleasing. 2) The results section can list sub-headings for each section separately to make the logic clearer.

Answer: Thank you. We formatted the Tables as suggested and in alignment with the journal’s requirements.

The Results section now has 3 subheadings to make it clearer: baseline characteristics, reliability, external validity. 

18. DISCUSSION/ Overall: 1) The discussion section has too few references, please add relevant arguments. 2) The discussion section did not explain the interpretation of the results and subsequent processing methods. Please fully explain and compare the results section.

Answer: Thank you for your valuable suggestions. We revised the Discussion section extensively, added more references to support relevant arguments. We also added more interpretations of the results and compared/contrasted findings with the literature. Kindly refer to the yellow-highlighted sentences in the revised Discussion section. 

19. CONCLUSIONS: Line 265-266, page 14: So what is the relationship between the quality of life of women in the early postpartum period?

Answer: Kindly note that the aim of our study was not to establish a relationship between QOL of postpartum women and breastfeeding or other variables. The aim was to explore its validity and reliability in our context. Satisfaction with breastfeeding as measured by the MBFES scores was used to establish the external validity of the MAPP-QOL. We have revised the conclusions for clarity as follows: 

“The Arabic MAPP-QOL has good psychometric properties. This instrument may be a useful tool for clinicians as well as researchers interested in measuring maternal postpartum QOL in the Arab contest. Further replication of our findings in other Arab contexts is needed.”

20. The starting point and research theme of this article are innovative. However, the logic and English formulation need to be strengthened.

Answer: Thank you. We revised the manuscript in light of all of your suggestions and comments, including the English language. We hope it meets your satisfaction now.

21. Why is confirmatory factor analysis not directly used for QOLs with relatively stable structures?

Answer: We addressed this comment above. Kindly refer to our reply to comment 2 and comment 12.

22. The discussion section did not explain the results reasonably and was confusing. It is recommended to read more references to provide theoretical support.

Answer: We revised the Discussion and provided more explanations of our findings with comparisons to the literature. Kindly refer to the yellow-highlighted sentences in the Discussion section. 

23. The typography of the text of the whole article is particularly messy, please pay attention to the alignment. 

Answer: Thank you for the alert. Done. 

Response to Reviewer 3:

Thank you for the valuable time you put to review our manuscript. Please find below our reply to each comment.

1. This is a very well-written and valuable paper. Some minor suggestions. 

Answer: Thank you very much. We greatly appreciate your feedback.

2. The method of sampling should be added in method portion of manuscript.

Answer: We added the following statements about the sampling method:

Abstract/Methods: “This instrument validation cohort study tested an adapted Arabic version of the MAPP-QOL questionnaire on a convenience sample of 485 healthy Lebanese postpartum women.”

Methods/Design and setting: “Participants were consecutively recruited from the postpartum ward of a tertiary care academic center in the capital city Beirut.”

3. The communalities of all items need to be provided alongside the factor loads.

Answer: We added the communalities of all items to Table 2. 

4. Please include the Arabic version of the MAPP-QOL as a supplementary material for the accessible use of other researchers and clinicians.

Answer: The Arabic MAPP-QOL is included as Supplementary file (S1 File). 

5. Why confirmatory factor analysis (CFA) has not been used?

Answer: Thank you for your comment. The reason we conducted Exploratory Factor Analysis is because the original English MAPP-QOL was validated in only 2 contexts (Brazilian and Persian) that are distinctly different from the Lebanese context. Hence, we wanted to explore how would it perform in our culture. Confirmatory Factor Analysis would be more suitable when the instrument has been well established to have good reliability and validity in most settings (e.g. The Beck Depression Inventory used to screen for depression and its severity in people over 13 years of age). 

6. Have the face validity and content validity been performed after the forward-backward translation?

Answer: Face and content validities were assessed by both authors before piloting. Both authors were in full agreement that the Arabic version had good face validity and content validity. The items in the questionnaire captured almost all facets of quality of life specific for the postpartum period such as pain, breasts, support, health issues, time for children, family, friends, employment, etc...Moreover, results of piloting the questionnaire on 20 women supported its face and content validities. 

7. Although the slope on the scree plot was leveling after fourth factor, it seems that the decision to retain five factors was to some extent arbitrary. Additional argument for such a decision could be obtained by employing some further procedures aimed at determining the optimal number of factors, such as Hornn’s parallel analysis or Velicer’s minimum average partial. 

Answer: We greatly appreciate your suggestion. The reason we chose five factors instead of 5 are based on the following reasons: 1) The Scree plot between the fourth and the fifth factor is going downward steeply whereas it is leveling after the fifth factor; 2) the total variance explained by 5 factors was 43.7% vs. 39.9% total variance explained by 4 factors.

As for your suggestion to conduct Hornn’s parallel analysis or Velicer’s minimum average partial, kindly note that we are not familiar with those tests. Hence, we consulted with our biostatistician and another epidemiologist who is an expert in Instrument validation. Both consultants stated that it is not a must to do these tests, and the biostatistician stated that he had not run such tests before. Any further advice on this issue from the kind reviewer is very much appreciated.

We hope we have addressed all the comments of the kind reviewers in a clear manner and revised the manuscript accordingly.

Thank you again for the thorough review of our paper.

Best wishes.

Sincerely, 

Mona Nabulsi, MD, MSc

Professor of Clinical Pediatrics

Department of Pediatrics and Adolescent Medicine

Faculty of Medicine

American University of Beirut

Beirut-Lebanon

P.O.Box: 113-6044/C8

E-mail: mn04@aub.edu.lb

---

## [Decision Letter · Decision Letter 1]

31 Jul 2023

PONE-D-23-06864R1Validation of the Arabic Maternal Postpartum Quality of Life Questionnaire Among Lebanese Women: A cohort studyPLOS ONE

Dear Dr. Nabulsi,<table border="0" cellpadding="0" cellspacing="0" class="datatable3" style="border-collapse: collapse; width: 678px; line-height: 14px; color: rgb(0, 0, 51); font-family: verdana, geneva, arial, helvetica, sans-serif; font-size: 11.2px;"> 

</table>Thank you for submitting your manuscript to PLOS ONE. After careful consideration, we feel that it has merit but does not fully meet PLOS ONE’s publication criteria as it currently stands. Therefore, we invite you to submit a revised version of the manuscript that addresses the points raised during the review process.

 Please submit your revised manuscript by Sep 14 2023 11:59PM. If you will need more time than this to complete your revisions, please reply to this message or contact the journal office at plosone@plos.org. Please include the following items when submitting your revised manuscript:A rebuttal letter that responds to each point raised by the academic editor and reviewer(s). You should upload this letter as a separate file labeled 'Response to Reviewers'.A marked-up copy of your manuscript that highlights changes made to the original version. You should upload this as a separate file labeled 'Revised Manuscript with Track Changes'.An unmarked version of your revised paper without tracked changes. You should upload this as a separate file labeled 'Manuscript'.If applicable, we recommend that you deposit your laboratory protocols in protocols.io to enhance the reproducibility of your results. Protocols.io assigns your protocol its own identifier (DOI) so that it can be cited independently in the future. For instructions see: https://journals.plos.org/plosone/s/submission-guidelines#loc-laboratory-protocols. Additionally, PLOS ONE offers an option for publishing peer-reviewed Lab Protocol articles, which describe protocols hosted on protocols.io. Read more information on sharing protocols at https://plos.org/protocols?utm_medium=editorial-email&utm_source=authorletters&utm_campaign=protocols.

We look forward to receiving your revised manuscript.

Kind regards,

Othman A. Alfuqaha, Ph.D.

Academic Editor

PLOS ONE

Journal Requirements:

Additional Editor Comments:

Abstract:

Repeated sentence in result section, please remove it. "All statistical analyses were done using SPSS version 23."]

Results:

In the validity section, Table 2, please add the total variation of the validated scale and eigenvalues for all dimensions.

Validity:

The results of the KMO and Bartlett test of sphericity should be presented in the validity section rather than the reliability section.

Discussion:

Additional references are needed in the discussion to compare the findings with previous studies. This allows for a broader understanding of the results and their implications.

Best of luck with your research!

Reviewers' comments:

Reviewer's Responses to Questions

**Comments to the Author**

1. If the authors have adequately addressed your comments raised in a previous round of review and you feel that this manuscript is now acceptable for publication, you may indicate that here to bypass the “Comments to the Author” section, enter your conflict of interest statement in the “Confidential to Editor” section, and submit your "Accept" recommendation.

Reviewer #2: (No Response)

2. Is the manuscript technically sound, and do the data support the conclusions?

Reviewer #2: Yes

3. Has the statistical analysis been performed appropriately and rigorously? 

Reviewer #2: Yes

4. Have the authors made all data underlying the findings in their manuscript fully available?

Reviewer #2: Yes

5. Is the manuscript presented in an intelligible fashion and written in standard English?

Reviewer #2: Yes

6. Review Comments to the Author

Reviewer #2: ID: PONE-D-23-06864R1

Title: Validation of the Arabic Maternal Postpartum Quality of Life Questionnaire Among Lebanese Women: A cohort study

Thank you for providing a chance to review this manuscript.

Comment: Minor Revision.

Detailed information:

Abstract

Results: 1) Please delete the sentence “All statistical analyses were done using SPSS version 23”.2) Please make the text more concise.

BACKGROUND

Overall: Why was this study conducted in the Arab region? Could some research background be added on the quality of life of postpartum women in Lebanon or Arabia?

Methods

Statistical Analysis

Pages 8-9, Line 172-179: After reading your reply, can you conduct confirmatory factor analysis on top of exploratory factor analysis? After all, exploratory factor analysis alone is not sufficient in the study of measurement characteristics of scales.

Results

Baseline characteristics

Page 9, Line 191-192: The language description is too concise. Do you want to display a flowchart? The flowchart should be presented in the methodology section, and the flowchart is too concise, like a draft.

Discussion

Paragraph 5, page 6: Please state the advantages of this study and the contribution to future research.

Thank you and my best,

Your reviewer

7. PLOS authors have the option to publish the peer review history of their article (what does this mean?). If published, this will include your full peer review and any attached files.

Reviewer #2: No

---

## [Author Response · Author response to Decision Letter 1]

5 Sep 2023

September 4, 2023 

Dear Editor,

We would like to submit our revised manuscript [PONE-D-23-06864] entitled “Validation of the Arabic Maternal Postpartum Quality of Life Questionnaire Among Lebanese women: A Cohort Study” for publication in PLOS ONE. All changes have been highlighted in yellow in the revised tracked copy. We are very thankful for the editor and the reviewers for the time they dedicated to review our manuscript and provide us with their valuable comments.

Below, please find our response to each comment.

Journal Requirements:

Answer: We reviewed our reference list meticulously and did not identify any article that was retracted. We would greatly appreciate specifying which article in our reference list the kind editor is referring to.

Additional Editor’s comments:

1. Authorship update: Kindly note that we added the PhD candidate Ms. Nour Abou Khalil as a third author to the list of authors in the revised manuscript. Dr. Abou Khalil conducted the Confirmatory Factor Analysis that was requested by Reviewer 2.

2. Abstract: Repeated sentence in Results section, please remove it. “All statistical analyses were done using SPSS version 23”.

Answer: Done. Thank you for alerting us to this error. 

3. Results: In the validity section, Table 2, please add the total variation of the validated scale and the eigen values for all dimensions.

Answer: We added the total variation of the scale and Eigen values of each component. They are summarized in a new table labelled as Table 2. The old Table 2 is now labelled as Table 3. We prefer not to merge the 2 tables but keep them separate for clarity.

Table 2. Eigen values for the 5 components of the scale and the total variance.

Component Initial Eigen Values

 Total % of Variance Cumulative %

1 8.570 21.975 21.975

2 2.878 7.380 29.355

3 1.998 5.123 34.477

4 1.807 4.633 39.111

5 1.374 3.523 42.633

4. Validity: The results of the KMO and Bartlett test of sphericity should be presented in the validity section rather than the reliability section.

Answer: We moved the sentence about the results of the KMO and Bartlett test of sphericity from the reliability section to the validity section next to the component analysis. 

5. Discussion: Additional references are needed in the Discussion to compare the findings with previous studies. This allows for a broader understanding of the results and their implications. 

Answer: In the 2nd paragraph of the Discussion, we compared and contrasted the results of our instrument (the Arabic MAPP-QOL) to the original English MAPP-QOL with respect to the subscales and relocation of items explaining the relocations in relation to our context. We also compared our finding of the association between better quality of life and the participants’ satisfaction with their overall breastfeeding experience, maternal enjoyment and role attainment, and lifestyle and body image to the literature (References 13, 17, 18, 19). Moreover, we discussed the differences between our instrument and the English, Persian, and Brazilian MAPP-QOL in terms of sample size and breastfeeding (in the Discussion). Kindly note that in the Introduction, we discussed in detail the psychometrics of the English, Brazilian, and Persian versions of the MAPP-QOL. We also discussed in the Introduction the existing literature on the QOL of postpartum women in the Arabic context. We feel that re-discussing the comparisons made in the Introduction again in the Discussion would be redundant. To our knowledge, these MAPP-QOL versions are the only ones in the literature. 

Response to Reviewer #2:

Thank you for the valuable time you put to review our manuscript. Please find below our reply to each comment.

1. Abstract: Results: 1) Please delete the sentence “All statistical analyses were done using SPSS version 23”. 2) Please make the text more concise.

Answer: We deleted the repeated sentence. We removed one long sentence from the Results to make the text more concise. After careful review of the remaining sentences, we felt that none of them could be deleted since they are summarizing the reliability and validity assessment of the scale. 

2. Background: Overall: Why was this study conducted in the Arab region? Could some research background be added on the quality of life of postpartum women in Lebanon or Arabia?

Answer: We added the below summary of the few available studies about QOL in postpartum Arab women to the Introduction:

“There have been no studies from Lebanon that described the quality of life in postpartum women. Few studies from the Arab world assessed the QOL of postpartum Arab women and its association with different health-related issues or conditions [9-11]. A study from Saudi Arabia reported that 59.68% of postpartum women had probable postpartum depression, with an inverse correlation between postpartum depression and health-related quality of life scores measured by the Health-Related Quality of Life Scale [9]. In another cross-sectional study, Badr et al. recruited a convenience sample of Arabic-speaking postpartum women through Arab groups on Facebook and WhatsApp. They examined the associations between postpartum fatigue, depressive cognitions, life satisfaction, resourcefulness, and quality of life using the single-item global QOL question [10]. This study found significant correlations between postpartum fatigue and each of the three remaining variables. A third study from Egypt [11] assessed the quality of life in postpartum women in relation to the various modes of delivery using the Chinese QOL by Zhou, et al., the reliability and validity of which were not tested [8]. There were no significant differences in the QOL aspects between mothers who delivered vaginally and those who delivered via Cesarean section. To note, none of the previous studies used the MAPP-QOL that is specific for the postpartum period. In view of the importance of the postpartum period in a woman’s life, and its associated physical, emotional, and social changes, there is a need for a valid instrument that can reliably assess the postpartum QOL in Arab women” 

3. Methods: Statistical Analysis, Pages 8-9, Line 172-179: After reading your reply, can you conduct confirmatory factor analysis on top of exploratory factor analysis? After all, exploratory factor analysis alone is not sufficient in the study of measurement characteristics of scales.

Answer: We appreciate the reviewer’s thoughtful question regarding our choice to conduct Exploratory Factor Analysis (EFA) rather than Confirmatory Factor Analysis (CFA) on the Arabic-PPQOL scale. Our decision to initially conduct EFA was driven by the unique context and translation of the scale. Given the cultural and contextual variations, we aimed to explore which factor structure would align optimally with our data. EFA allowed us to identify underlying patterns and relationships among items without imposing any preconceived factor structure. Subsequently, we performed CFA to assess the adequacy of the hypothesized factor structure proposed by Hill et al. (2006) for our dataset as per your suggestion. The CFA results indicated that the model reasonably fits the data without a perfect match given the unique context of our study. We added the following paragraph to the Methods, statistical analysis section:

“Moreover, a confirmatory factor Analysis (CFA) was conducted using STATA 14 software to verify the factor structure for MAPP-QOL hypothesized by Hill et al. [5]. The maximum-likelihood estimation method was employed, and a range of widely recognized goodness of fit indices were utilized to assess the adequacy of the model. The model fitness was assessed based on the Root Mean Square of Error of Approximation (RMSEA), Comparative Fit Index (CFI), Standardized Root Mean Square Residual (SRMR), and Chi-Square Statistic”. 

We also added the following paragraph to the Results section:

“In the CFA, after modification of indices, the goodness of fit indices corresponding to the 5-factor model suggested in the original questionnaire indicated a reasonable fit with RMSEA= 0.052, CFI=0.847 and SRMR=0.062. We failed to reproduce the covariance matrix with a p-value <0.0001. Additionally, all items showed significant factor loadings on their respective factors. The lowest factor loading was observed for item 8 “Your surgical incision or episiotomy”, whereas the highest was observed for item 30 “Your ability to meet financial obligations”. 

In the Discussion, we added the following statement:

“In our attempt to examine the fit of the original factor structure suggested by Hill et al. [5] using CFA, the model seemed to approximate the data reasonably, yet it did not demonstrate perfect alignment with our observed data given the unique context of our study”.

4. Results: Baseline characteristics, Page 9, Line 191-192: The language description is too concise. Do you want to display a flowchart? The flowchart should be presented in the methodology section, and the flowchart is too concise, like a draft.

Answer: Kindly note that Figure 1 displays a flowchart of the participants’ progress through the study. We also revised this sentence as follows: 

“We recruited a total of 485 participants, of whom 24 withdrew by the second week, 11 withdrew between two weeks and one month, and 2 withdrew between one and three months, with an overall loss to follow up of 7.6% (Fig 1)” 

5. Discussion: Paragraph 5, page 6: Please state the advantage of this study and the contribution to future research.

Answer: We added the following sentences to the Conclusion:

“This study provides the first Arabic instrument that can be used by clinicians to measure the QOL of Arab postpartum women, a period with relatively increased prevalence of significant social stressors and physical ailments that can negatively affect their QOL. Moreover, this valid instrument may bolster research efforts focusing on QOL issues in the Arab world. Future research is needed to replicate our findings in other Arab contexts, as well as to further refine the instrument for optimal fit with our cultural and contextual variations.” 

We hope we have addressed all the comments of the kind reviewers in a clear manner and revised the manuscript accordingly.

Thank you again for the thorough review of our paper.

Best wishes.

Sincerely, 

Mona Nabulsi, MD, MSc

Professor of Clinical Pediatrics

Department of Pediatrics and Adolescent Medicine

Faculty of Medicine

American University of Beirut

Beirut-Lebanon

P.O.Box: 113-6044/C8

E-mail: mn04@aub.edu.lb

---

## [Editor Report · Decision Letter 2]

6 Sep 2023

Validation of the Arabic Maternal Postpartum Quality of Life Questionnaire Among Lebanese Women: A cohort study

PONE-D-23-06864R2

Dear Dr. Mona Nabulsi,

We’re pleased to inform you that your manuscript has been judged scientifically suitable for publication and will be formally accepted for publication once it meets all outstanding technical requirements.

Kind regards,

Othman A. Alfuqaha, Ph.D.

Academic Editor

PLOS ONE

Additional Editor Comments (optional):

Congratulations on successfully addressing all of my comments and reviewers' comments. I would like to take this moment to express my gratitude for your effort in creating this amazing article. Best of luck, and may all your endeavors be met with success. Wishing you the very best in all your future endeavors.
---

## [Editor Report · Acceptance letter]

19 Sep 2023

PONE-D-23-06864R2 

Validation of the Arabic Maternal Postpartum Quality of Life Questionnaire Among Lebanese Women: A cohort study 

Dear Dr. Nabulsi:

I'm pleased to inform you that your manuscript has been deemed suitable for publication in PLOS ONE. Congratulations! Your manuscript is now with our production department. 

Kind regards, 

on behalf of

Dr. Othman A. Alfuqaha 

Academic Editor

PLOS ONE